# TEFORMER: A TOPOLOGY-ENHANCED TRANSFORMER FOR ARCHITECTURE PERFORMANCE PREDICTION

## ABSTRACT

Evaluating architecture performance is a crucial step in neural architecture search (NAS) but remains computationally expensive. Performance predictors offer an efficient alternative by learning from a limited set of architecture-performance pairs. However, previous predictors tend to oversimplify the topological structure of neural architectures using adjacency matrices, node depths, or computation flow, which fail to fully capture topological features of architectures, leading to poor generalization. To address this limitation, we propose TEFormer, a Topology-Enhanced Transformer that integrates both local and global topological information beneficial to performance prediction. Specifically, we employ a topology-aware flow encoding module that incorporates local topological characteristics via a learnable structural encoding and a flow-based encoder. At the global level, we design a hierarchical attention mechanism to jointly model intra-flow and inter-flow interactions within the architecture. To further improve generalization, we propose an architecture augmentation strategy that synthesizes additional samples by interpolating similar architectures in the latent space. Extensive experiments on computer vision, graph learning, and automatic speech recognition tasks demonstrate that TEFormer consistently outperforms state-of-the-art predictors and exhibits superb performance across diverse search spaces.

## 1 INTRODUCTION

Neural architecture search (NAS) provides an automated solution for discovering well-performing architectures, achieving superior results over handcrafted models on various domains (Cai et al., 2020; Ghiasi et al., 2019; Wang et al., 2020; Gao et al., 2020). Typically, NAS consists of three key components: the search space, the search strategy, and the performance evaluation (Elsken et al., 2019). Among these, accurate and efficient performance evaluation is crucial for reducing the cost of NAS. Conventional NAS evaluates candidate architectures by training them from scratch (Zoph & Le, 2016; Real et al., 2017), which is notoriously time-consuming. As an alternative, performance predictors (Luo et al., 2018; Ning et al., 2020) can estimate the performance of unseen architectures once trained with a small set of labeled samples, significantly accelerating the NAS process. Since the topological structure of an architecture plays an essential role in its performance, how to effectively model such topology becomes a core challenge in performance predictors.

Prevalent methods (Wen et al., 2020; Dudziak et al., 2020) employ graph neural networks (GNNs) to encode architecture topology through adjacency matrices. Recent works (Yi et al., 2023a;b) adopt Transformers that treat architectures as fully-connected graphs, incorporating depth-based position encoding and multi-head self-attention to capture long-range dependencies between operations.

Despite the success of Transformer-based predictors, they tend to overlook the topological order within the architecture where operations are processed sequentially. In practice, this order corresponds to the forward and backward computation flows of the architecture, functioning as a critical cue for performance. To incorporate such topological cues, flow-based predictors (Ning et al., 2020; 2022; Hwang et al., 2024) explicitly model the computation flow via asynchronous message passing. Each operation is updated only after all its predecessors (i.e., operations with incoming edges) have been processed, thus preserving flow-specific topological information.

However, existing flow-based approaches still face two key limitations in topological modeling. First, they lack explicit structural encoding, struggling to distinguish operations with similar type but different topological properties. In these predictors, an operation is updated based solely on the embedding of its predecessors, without considering topological properties such as relative distance or degree of connectivity. As a result, important patterns of architectures, such as skip connections (He et al., 2016), may not be adequately captured because all operations with the same type are treated identically during the message passing regardless of their positions. Second, global interactions are restricted to individual flows. For instance, GATES (Ning et al., 2020) and TA-GATES (Ning et al., 2022) constrain information propagation to directly connected operations, while FlowerFormer (Hwang et al., 2024) only models the mutual dependencies along a single flow. In practice, an architecture often employs multiple concurrent flows to process input data as different flows can focus on different characteristics of data. For example, in the Inception module (Szegedy et al., 2015; 2016), parallel convolutions with varying kernel sizes extract multi-scale features of data that are complementary to each other. Therefore, capturing such cross-flow interactions is crucial for precise performance estimation.

To overcome these limitations, we propose a Topology-Enhanced Transformer (TEFormer) that fully exploits both local and global topological information of neural architectures. Specifically, we first introduce a topology-aware flow encoding module that embeds local topological features into the modeling of bidirectional computation flow via a learnable structural encoding. The structural encoding is initialized with random walk probabilities, which can express the relative distances (Ying et al., 2021) and high-order neighborhoods of operations (Gasteiger et al., 2019). Given the nature of random walks and the bidirectional flow modeling, this approach is particularly well-suited for capturing the complex directional dependencies in neural architectures compared with popular position encodings like shortest relative distance and Laplacian eigenvector. Besides, we design a hierarchical attention mechanism that computes attention scores over a masked adjacent matrix. This enables TEFormer to capture intra-flow and inter-flow dependencies among operations globally.

Due to the high cost of obtaining ground-truth performance of architectures, only a small number of labeled architectures are available for training performance predictors, highlighting the need for strong generalization of predictors under data-scarce conditions. Graph Transformers are particularly prone to overfitting in such few-shot scenarios (Ma et al., 2023). Although prior works (Liu et al., 2021; Yi et al., 2023a) augment data by permuting operation orders of architectures, they are incompatible with our topology-aware flow encoding module, which processes operations strictly according to the topological order of the architecture regardless of the permutations. To address this, we propose an interpolation-based architecture augmentation strategy. For each architecture, we find its most similar counterpart within a mini-batch based on representation and label similarity, and interpolate their latent representations to synthesize augmented samples. This interpolation-based strategy not only improves the generalization of TEFormer in low-data regimes but also helps ensure the validity of generated samples, as directly operating on raw architectures may yield invalid ones, such as architectures with incorrect computation flows.

We conduct rigorous experiments across a wide range of NAS search spaces, covering computer vision, graph learning, and automated speech recognition tasks. Experimental results demonstrate the effectiveness of TEFormer in evaluating architecture performance and searching for promising architectures. For instance, TEFormer can discover well-performing architectures with 76.4% and 97.59% classification accuracy on ImageNet and CIFAR-10 datasets, respectively. Our key contributions are summarized as follows:

- We introduce TEFormer, a Topology-Enhanced Transformer that integrates local topology-aware flow encoding and global hierarchical attention to capture complex architecture topology for accurate performance prediction.

- We design a novel interpolation-based architecture augmentation strategy that synthesizes informative training samples, fueling TEFormer to generalize well in data-scarce scenarios.

- We conduct extensive experiments on diverse NAS search spaces, showing that TEFormer consistently outperforms competitive predictors. Ablation studies and sensitivity analyses further validate the contribution of each component.

## 2 RELATED WORKS

### 2.1 PERFORMANCE PREDICTOR FOR NAS

Performance predictors aim to directly estimate unseen architectures after being trained on a small number of labeled samples, thus significantly improving the evaluation efficiency of NAS. Typically, a predictor is composed of an encoder that transforms input architectures into informative representations and a regressor that predicts corresponding performance. Extracting topological features of architectures is a critical challenge for predictors. Early approaches adopt multi-layer perceptron (MLP) as the encoder (Cai et al., 2020; Liu et al., 2018), treating each architecture as a fixed-length sequence. However, this paradigm often performs poorly due to inappropriate structural encoding of architectures. To harness rich topological information, a large number of predictors embrace GNN and Transformer as encoders to model architectures as graphs. For instance, the GNN-based BRP-NAS (Dudziak et al., 2020) and the Transformer-based NAR-Former (Yi et al., 2023a) employ adjacency matrix and depth-based positional encoding to model the topological structure. Despite their effectiveness, these methods still ignore the computation flow within the architecture (i.e., forward and backward passes), which conveys vital topological cues for performance prediction.

To preserve the flow information, flow-based methods process operations following the topological order inherent in the architecture. GATES (Ning et al., 2020) and TA-GATES (Ning et al., 2022) represent architectures as directed acyclic graphs (DAGs) and update operation embeddings in the same order that information propagates through the architecture. Nevertheless, both predictors are limited to local structural patterns. FlowerFormer (Hwang et al., 2024) extends this line by incorporating global interactions through a flow-aware attention module, resulting in notable performance improvements. Nevertheless, it does not integrate any structure encoding into the architecture representation, which inevitably leads to the loss of topological information (Zhu et al., 2023).

### 2.2 GRAPH TRANSFORMERS

Transformers (Vaswani et al., 2017) have achieved remarkable success in natural language processing and demonstrated great potential in graph learning because of their powerful self-attention mechanism. Compared to sequences, graph topology is more complex, making it crucial to inject topological information of graphs into Transformers. A common scheme enriches graph representations with various structural encodings, such as Laplacian eigenvectors (Kreuzer et al., 2021), shortest relative distance (Ying et al., 2021), and random walks (Ma et al., 2023). In this paper, we adopt a learnable structural encoding based on random walks because it can effectively express multi-hop dependencies and directional patterns that are crucial for modeling DAG-like neural architectures.

Another line of approaches revolves around tailoring attention mechanisms to model graph topology. For instance, SAT (Chen et al., 2022) integrates subgraph information into attention computation, and Exphormer (Shirzad et al., 2023) introduces virtual nodes to approximate fully connected graphs. DAGFormer (Luo et al., 2023) highlights the structural bias in directed graphs by restricting the attention of each node to its predecessors and successors along directed paths. However, these methods are proposed for generic graph learning and often require domain-specific adaptations (Luo et al., 2023) like neural architecture modeling. To this end, we propose a hierarchical attention mechanism that captures intra-flow and inter-flow dependencies to refine architecture representations.

## 3 PRELIMINARY

**Notation.** We denote each input architecture $X = (\mathcal{V}, \mathcal{E})$ as a DAG, where the node set $\mathcal{V}$ stands for operations and the edge set $\mathcal{E}$ represents the connections. Let $A \in \{0, 1\}^{N \times N}$ denote the asymmetric adjacency matrix of an architecture with $N$ nodes, where $A_{u,v} = 1$ when $(u, v) \in \mathcal{E}$. $A^\top$ is the transpose of $A$. If $(u, v) \in \mathcal{E}$, we call $u$ a predecessor of $v$ and use $P(v)$ as the predecessors of $v$. We denote the degree, in-degree and out-degree matrix as $D, D_{\text{in}}, D_{\text{out}} \in \mathbb{R}^{N \times N}$, respectively.

**Random Walks on Graphs.** The topological relationship between nodes can be expressed by a $k$-step random walk. A transition matrix is defined by $T = D^{-1}A \in [0, 1]^{N \times N}$, where $T_{u,v}$ represents the probability of transitioning node $u$ to node $v$ in a single random walk step (Li et al., 2020). The $k$-step random walk matrix is then computed as $RW = \{T, T^2, \dots, T^k\} \in [0, 1]^{k \times N \times N}$.

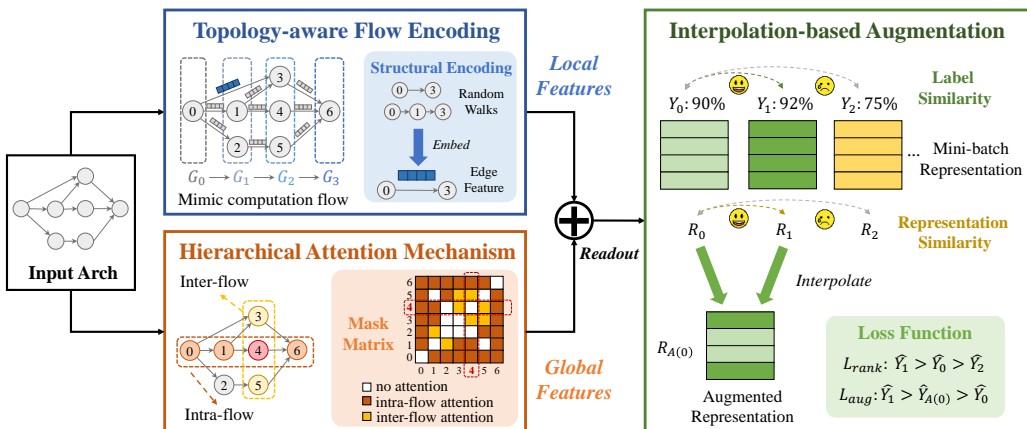

Figure 1: **Overview of TEFormer**. TEFormer extracts local and global topological features through two dedicated modules. For local features, the topology-aware flow encoding module embeds topological information into the flow-based encoder via a learnable structural encoding based on random walks. The operations are updated according to a fixed topological order. For global features, the proposed hierarchical attention mechanism models inter-flow and intra-flow dependencies through a tailored a mask matrix. Subsequently, both features are aggregated to obtain the architecture representation $R$ through a readout function. Finally, we design an augmentation strategy to enrich the training set, which interpolates each architecture with the most similar one in terms of representation and label within the mini-batch. The overall loss function includes a ranking loss $L_{rank}$ that encourages correct relative orderings, and an augmentation loss $L_{aug}$ that places the performance of augmented samples between the source architectures.

## 4 METHODOLOGY

In this section, we introduce TEFormer, a Transformer that effectively captures both local and global topological information of neural architectures. We begin by detailing the topology-aware flow encoding module that injects topological characteristics into the flow encoding. Next, we present a hierarchical attention mechanism that models the dependencies along and across multiple flows. Finally, we propose an interpolation-based architecture augmentation strategy to mitigate the overfitting problem of the predictor. The overall framework of TEFormer is illustrated in Figure 1.

### 4.1 TOPOLOGY-AWARE FLOW ENCODING MODULE

We follow the common flow-based scheme (Ning et al., 2022; Hwang et al., 2024) to update operations based on the topological order defined by the architecture. To encode local topological information, we introduce a learnable structural encoding. We start by describing how operations are partitioned into multiple groups.

**Topological Group Partition.** For an architecture $X = (\mathcal{V}, \mathcal{E})$, operations are divided into $B$ topological groups $G_0, G_1, \ldots, G_B$, where each group is processed only after earlier groups are updated. The first group $G_0$ is composed of input nodes. After that, at each step $t$, we create a new group $G_t$ by scanning all the unassigned nodes. A node is added to group $G_t$ if all the nodes it depends on have already been assigned to earlier groups $G_0, G_1, \ldots, G_{t-1}$:

$$G_t = \left\{ v \in \mathcal{V} \setminus \bigcup_{i=0}^{t-1} G_i \,\middle|\, \forall (u,v) \in \mathcal{E}, \ u \in \bigcup_{i=0}^{t-1} G_i \right\} \tag{1}$$

This ensures that any node in group $G_t$ can use all the updated information it needs from prior groups. This process is iterative and continues until all nodes have been assigned. The final group contains only the output node.

**Learnable Structural Encoding.** Once the topological groups are determined, we update node features strictly following the order of the assigned groups. For each node $u$, its feature $h_u$ is updated by aggregating messages from its predecessors and combining these messages with its own feature. To preserve topological information during the aggregation process, we incorporate a learnable structural encoding based on $k$-step random walks $RW = \{T, T^2, \ldots, T^k\}$ as introduced in the preliminary, where $k$ is set to the number of topological groups. Each element $RW_{v,u}$ is passed through an MLP to generate the learnable structural encoding $SE_{v,u}$. This encoding is used to initialize the corresponding edge feature $e_{v,u}$, thus expressing the relative relationship.

Considering that both forward and backward passes are included when modeling neural architectures in flow-based methods (Ning et al., 2022; Hwang et al., 2024), we follow this convention and encode the bidirectional computation flow.

**Forward Flow Encoding.** In the $l$-th flow layer, each node updates its hidden state by combining its previous state with the information propagated from its predecessor nodes. The message from the predecessor $v$ is generated by assigning an attention weight based on the predecessor's feature $h_v^l$, the previous feature of target node $h_u^{l-1}$, and the corresponding edge feature $e_{v,u}^l$:

$$h_{v,u}^l = \text{Softmax}(w_1^{l\top} h_v^l + w_2^{l\top} h_u^{l-1} + w_3^{l\top} e_{v,u}^l)\, h_v^l, \quad v \in P(u) \tag{2}$$

where $w_1^l$, $w_2^l$, and $w_3^l$ are trainable weights. We than aggregate all predecessor messages to obtain the overall incoming information for node $u$. The aggregated message $h_{P(u)}^l$ can be computed as:

$$h_{P(u)}^l = \text{Agg}(\{h_{v,u}^l | v \in P(u)\}) \tag{3}$$

where Agg is an aggregate function. Here, we choose the addition function. Subsequently, we can update the state of node $u$ by combining its previous feature $h_u^{l-1}$ and the aggregated message from all predecessors $h_{P(u)}^l$:

$$h_u^l = \text{GRU}(h_u^{l-1}, h_{P(u)}^l) \tag{4}$$

where GRU denotes Gated Recurrent Unit (Thost & Chen, 2021), which functions as a combine operator and can model sequential data effectively. For input nodes, the initial state of the GRU is set to zero.

**Backward Flow Encoding.** After the forward pass, we incorporate the backward information by traversing the architecture in a reverse topological order, which starts from the output node. The reverse random walk is defined as $\tilde{RW} = \{\tilde{T}, \tilde{T}^2, \ldots, \tilde{T}^k\}$, where $\tilde{T} = D_{\text{in}}^{-1} A^\top$. Other updating steps are consistent with the forward process.

After modeling the bidirectional flows, we can obtain the output node feature $H_f^l \in \mathbb{R}^{N \times d}$ of the $l$-th flow layer, where $d$ is the feature dimension.

## 4.2 HIERARCHICAL ATTENTION MECHANISM

To leverage global topological features of architectures, we propose a hierarchical attention mechanism that focuses on long-range relationships along the individual flow and across different flows.

Unlike the standard self-attention that connects all pairs of nodes indiscriminately, we customize a binary mask matrix $M \in \{0, 1\}^{N \times N}$ to selectively involve the intra-flow and inter-flow interactions. For nodes $v$ and $u$, $M_{v,u}$ will be 1 if and only if 1) there exists a directed path between $v$ and $u$, or 2) $v$ and $u$ belong to the same topological group. The first type of mask expands the receptive field of each node from direct neighbors to all reachable nodes along a directed flow, thereby excavating richer computational dependencies of architectures. The second type of mask enables communication among nodes within the same topological group, which are executed in parallel in a real-world architecture, thus enabling effective information exchange across multiple computation flows.

Given the mask matrix $M$, we can obtain the node feature $H_{attn}^l \in \mathbb{R}^{N \times d}$ in the $l$-th attention layer:

$$H_{head-i}^l = (M \odot \text{Softmax}(\frac{Q_i^l (K_i^l)^\top}{\sqrt{d}}))\, V_i^l, \tag{5}$$

$$H_{attn}^l = \text{FeedForward}(\|_{i=1}^m H_{head-i}^l) \tag{6}$$

where $Q_i^l = H^{l-1}(W_Q)_i^l$, $K_i^l = H^{l-1}(W_K)_i^l$, and $V_i^l = H^{l-1}(W_V)_i^l$ denote the query, key, and value matrices of the $i$-th attention head, respectively. $H^{l-1}$ is the node feature of the $(l-1)$-th encoder layer. $(W_Q)_i^l$, $(W_K)_i^l$, and $(W_Q)_i^l$ are trainable parameters. $||$ denotes concatenation, $\odot$ denotes element-wise multiplication, $m$ is the number of attention heads.

The output node feature $H^l$ of the $l$-th encoder layer is obtained by fusing the local feature $H_f^l$ produced by the $l$-th flow layer and the global feature $H_{attn}^l$ produced by the $l$-th attention layer:

$$H^l = \text{BN}\left(\text{MLP}\left(H_f^l + H_{attn}^l\right) + H_f^l + H_{attn}^l\right), \tag{7}$$

where BN denotes batch normalization. In our implementation, MLP is a two-layer MLP with ReLU activation function. After that, we can generate the representation $R_X$ of architecture $X$ from the node feature $H^L \in \mathbb{R}^{N \times d}$ that output by the $L$-th encoder layer:

$$R_X = \text{Readout}(\{H_i^L | i \in \mathcal{V}\}) \tag{8}$$

where Readout is a graph pooling function. In our case, we adopt the simple average pooling.

### 4.3 INTERPOLATION-BASED ARCHITECTURE AUGMENTATION

The effectiveness of performance predictors heavily relies on the number of available labeled architectures, which are typically scarce due to the prohibitive cost of training. This scarcity of training data significantly hinders the generalization of predictors. To address this challenge, we introduce an interpolation-based augmentation strategy to enrich the training set and enhance generalization.

Specifically, we sample architecture pairs and perform interpolation in the latent space to create augmented samples. To ensure the quality of the augmented data (Wang et al., 2021b; Ling et al., 2023), each architecture is paired with another one that is similar in both representation space and performance label. The similarity between two architecture $(X_i, X_j)$ is defined as:

$$\text{Sim}(X_i, X_j) = \underbrace{\text{Norm}(R_i^\top R_j)}_{\text{Representation Similarity}} \cdot \underbrace{\exp(-|Y_i - Y_j|)}_{\text{Label Similarity}} \tag{9}$$

where $R_i$ and $R_j$ denote the representations of $X_i$ and $X_j$, respectively. $Y_i$ and $Y_j$ denote their corresponding ground-truth performance. Norm represents a min-max normalization operation. Both representation-level and label-level similarities are scaled to the range $(0, 1)$ and have equal importance in the overall similarity. For each architecture $X$, we select its most similar counterpart $X'$ within the mini-batch to perform interpolation. The resulting augmented representation $R_{A(X)}$ and its corresponding label $Y_{A(X)}$ are given by:

$$R_{A(X)} = \lambda R_X + (1 - \lambda)R_{X'}, Y_{A(X)} = \lambda Y_X + (1 - \lambda)Y_{X'}$$
$$\text{s.t.} \quad X' = \arg\max_{X'} \text{Sim}(X, X') \tag{10}$$

where $\lambda \sim \text{Beta}(\alpha, \alpha)$ and $\alpha$ is a hyperparameter controlling the degree of interpolation.

### 4.4 LOSS FUNCTION

To predict architecture performance from the learned representation, we place an MLP after the encoder. Since distinguishing relative ranking between architectures is more important than estimating their absolute performance (Xu et al., 2021), we employ a hinge ranking loss here:

$$\mathcal{L}_{rank} = \sum_{(i,j) \in S} [b - (\hat{Y}_i - \hat{Y}_j) * \text{sign}(Y_i - Y_j)]_+ \tag{11}$$

where $[\cdot]_+ = \max(0, \cdot)$. $S$ denotes the training set and $b$ is a margin parameter. $\mathcal{L}_{rank}$ will be 0 when the predicted ordering matches the ground truth and the prediction difference exceeds $b$.

Since each augmented sample is generated via interpolation, it is expected to exhibit intermediate performance between the source architectures. Hence, we design a loss function for augmentation:

$$\mathcal{L}_{aug} = \sum_{i \in S} [b - (\hat{Y}_i - \hat{Y}_{A(i)}) * \text{sign}(Y_i - Y_{A(i)})]_+ + [b - (\hat{Y}_i' - \hat{Y}_{A(i)}) * \text{sign}(Y_i' - Y_{A(i)})]_+ \tag{12}$$

where $Y_i'$ and $Y_{A(i)}$ denote the ground-truth performance of the paired sample and the interpolated performance of the augmented sample, and $\hat{Y}_i'$, $\hat{Y}_{A(i)}$ are their corresponding predicted scores.

Table 1: Kendall's Tau on NAS-Bench-101, NAS-Bench-201, and NAS-Bench-301. The results are scaled up by a factor of 100. **Bold** indicates the best.

| Search spaces | NAS-Bench-101 | | | | NAS-Bench-201 | | | | NAS-Bench-301 | | | |
|---|---|---|---|---|---|---|---|---|---|---|---|---|
| Training portions | 1% | 5% | 10% | 50% | 1% | 5% | 10% | 50% | 1% | 5% | 10% | 50% |
| GatedGCN (Bresson & Laurent, 2017) | 67.4 (6.0) | 79.6 (4.1) | 82.0 (5.1) | 84.8 (5.9) | 70.9 (1.8) | 84.1 (0.6) | 88.6 (0.3) | 92.3 (0.1) | 61.8 (2.4) | 70.0 (0.9) | 71.4 (1.0) | 72.7 (1.5) |
| DAGNN (Thost & Chen, 2021) | 72.4 (4.5) | 82.9 (3.1) | 84.4 (4.4) | 85.9 (5.3) | 75.8 (1.0) | 87.5 (0.8) | 90.6 (0.2) | 92.6 (0.0) | 61.5 (1.9) | 70.9 (0.5) | 73.4 (1.2) | 76.1 (1.3) |
| GraphGPS (Rampášek et al., 2022) | 70.6 (4.4) | 81.7 (3.8) | 83.9 (4.2) | 85.9 (5.1) | 71.3 (1.3) | 82.5 (0.6) | 87.8 (0.5) | 92.7 (0.1) | 59.7 (1.8) | 69.3 (0.9) | 70.7 (1.2) | 73.8 (0.7) |
| NAR-Former (Yi et al., 2023a) | 59.4 (8.8) | 72.0 (8.2) | 75.5 (10.2) | 79.8 (5.9) | 62.3 (4.0) | 80.7 (1.8) | 87.3 (0.7) | 88.9 (0.3) | - | - | - | - |
| TA-GATES (Ning et al., 2022) | 70.8 (6.0) | 82.3 (2.7) | 83.9 (3.5) | 86.3 (3.9) | 77.7 (1.7) | 86.3 (0.8) | 88.7 (0.3) | 91.4 (0.5) | 61.3 (1.2) | 68.9 (1.6) | 71.8 (1.6) | 75.4 (0.7) |
| FlowerFormer (Hwang et al., 2024) | 75.0 (2.9) | 86.1 (0.8) | 88.1 (0.2) | 89.6 (0.1) | 80.0 (0.8) | 89.8 (0.3) | 91.3 (0.2) | 92.9 (0.1) | 64.2 (1.6) | 72.2 (1.0) | 73.6 (1.3) | 77.5 (0.7) |
| TEFormer (ours) | **78.9 (3.8)** | **86.5 (0.2)** | **88.6 (0.1)** | **90.3 (0.1)** | **81.1 (0.4)** | **90.7 (0.2)** | **91.7 (0.1)** | **93.5 (0.1)** | **66.9 (1.1)** | **73.0 (0.5)** | **74.3 (0.5)** | **78.1 (0.1)** |

Table 2: Precision@K on NAS-Bench-101, NAS-Bench-201, and NAS-Bench-301. The training portion is set to 5%. The results are scaled up by a factor of 100. **Bold** indicates the best and underline indicates the second.

| Search spaces | NAS-Bench-101 | | | | NAS-Bench-201 | | | | NAS-Bench-301 | | | |
|---|---|---|---|---|---|---|---|---|---|---|---|---|
| K | 1 | 5 | 10 | 50 | 1 | 5 | 10 | 50 | 1 | 5 | 10 | 50 |
| GatedGCN (Bresson & Laurent, 2017) | 44.4 (7.6) | 65.6 (3.5) | 76.2 (2.7) | 90.5 (2.1) | 42.3 (3.7) | 68.5 (3.1) | 80.9 (1.9) | 94.1 (0.6) | 19.1 (4.1) | 55.2 (4.5) | 71.8 (2.9) | 85.4 (0.4) |
| DAGNN (Thost & Chen, 2021) | 41.7 (5.9) | 65.4 (4.2) | 79.3 (2.9) | 92.0 (1.3) | 49.6 (6.2) | 69.7 (3.0) | 83.1 (0.7) | 95.3 (0.9) | 23.1 (2.1) | 58.3 (3.4) | 73.1 (1.5) | 85.8 (0.4) |
| GraphGPS (Rampášek et al., 2022) | 44.3 (12.2) | 67.1 (2.7) | 78.7 (1.9) | 91.2 (2.0) | 49.4 (4.6) | 67.9 (4.9) | 78.9 (3.4) | 93.4 (0.3) | 20.6 (2.1) | 57.2 (3.8) | 73.4 (2.5) | 84.8 (0.5) |
| NAR-Former (Yi et al., 2023a) | 47.2 (9.9) | 62.6 (7.9) | 67.8 (8.4) | 85.9 (5.2) | 49.5 (6.5) | 64.7 (2.0) | 69.9 (2.0) | 92.3 (1.0) | - | - | - | - |
| TA-GATES (Ning et al., 2022) | 44.6 (9.7) | 66.6 (4.0) | 78.1 (4.6) | 91.8 (1.2) | 49.4 (3.1) | 66.7 (3.3) | 78.1 (2.8) | 94.8 (0.7) | 20.1 (5.0) | 56.2 (6.1) | 72.4 (3.6) | 84.7 (0.7) |
| FlowerFormer (Hwang et al., 2024) | 46.5 (11.2) | 70.0 (1.5) | 80.9 (1.8) | 92.7 (1.7) | 57.0 (5.4) | 74.7 (1.6) | 85.2 (1.3) | 96.9 (0.7) | 20.8 (3.7) | 58.5 (2.5) | 74.7 (2.4) | 86.6 (0.6) |
| TEFormer (ours) | **49.8 (7.6)** | 68.8 (1.1) | **81.9 (2.5)** | **93.1 (1.0)** | **58.7 (5.2)** | **75.5 (2.0)** | **86.5 (1.8)** | **97.4 (0.3)** | **25.4 (1.3)** | **65.2 (0.9)** | **78.6 (0.5)** | **86.9 (0.2)** |

Finally, the overall training objective combines both losses:

$$\mathcal{L} = \mathcal{L}_{rank} + \lambda_1 \mathcal{L}_{aug} \tag{13}$$

where $\lambda_1$ is a weighting coefficient that controls the intensity of the augmentation.

## 5 EXPERIMENTS

In Sec 5.1, we evaluate the ranking performance of TEFormer across five NAS search spaces. In Sec 5.2, we conduct search experiments on ImageNet (Deng et al., 2009) and CIFAR-10 (Krizhevsky et al., 2009) datasets. In Sec 5.3, we present ablation studies and sensitivity analyses for TEFormer. Results about computational efficiency can be found in Appendix E.1.

**Experimental Settings:** In ranking experiments, we compare TEFormer with competitive predictors on five search spaces, including computer vision (i.e., NAS-Bench-101 (Ying et al., 2019), NAS-Bench-201 (Dong & Yang, 2020), NAS-Bench-301 (Zela et al., 2022)), graph learning (i.e., NAS-Bench-Graph (Qin et al., 2022)), and automatic speech recognition tasks (i.e., NAS-Bench-ASR (Mehrotra et al., 2021)). For searching experiments, we use DARTS (Liu et al., 2019) space to search promising architectures on ImageNet (Deng et al., 2009) and CIFAR-10 (Krizhevsky et al., 2009). In parallel to prior works (Ning et al., 2020; 2022; Yi et al., 2023a), we mainly use Kendall's Tau (Sen, 1968) to measure the ranking performance of predictors. Besides, we adopt another popular metric, Precision@K (Hwang et al., 2024) (the portion of actual top-K% architectures among the predicted top-K% ones), to evaluate the top-ranking ability of predictors.

### 5.1 RANKING EXPERIMENTS

In this series of experiments, we follow the settings of data splits in the previous work (Hwang et al., 2024). We perform the experiments with various portions of the training set (1%, 5%, 10%, and 50%) and use another 40 architectures as the validation set. Results are averaged over nine runs.

**Results on Computer Vision Tasks:** We report the Kendall's Tau of TEFormer on three computer vision search spaces and compare it with several baseline methods in Table 1. It can be observed that TEFormer surpasses all competitors consistently across all search spaces. Considering that only a small fraction of the entire search space can be used for training the predictor in practice, we primarily focus on the results under low training data. Compared to the previous state-of-the-art FlowerFormer (Hwang et al., 2024), TEFormer improves Kendall's Tau by 3.9 on NAS-Bench-

Table 3: Kendall's Tau on NAS-Bench-Graph (NB-G) and NAS-Bench-ASR (NB-ASR). The results are scaled up by a factor of 100. **Bold** indicates the best.

| Search Spaces | Predictors | Training portions | | | |
|---|---|---|---|---|---|
| | | 1% | 5% | 10% | 50% |
| NB-G | DAGNN (Thost & Chen, 2021) | 48.1 (3.2) | 64.4 (1.2) | 67.4 (1.1) | 73.1 (0.8) |
| | TA-GATES (Ning et al., 2022) | 33.1 (1.4) | 34.1 (2.0) | 35.4 (0.8) | 35.7 (0.5) |
| | FlowerFormer (Hwang et al., 2024) | 49.5 (1.1) | 65.9 (1.3) | 68.9 (0.6) | 72.7 (0.2) |
| | TEFormer (ours) | **52.0** (1.1) | **67.1** (0.4) | **69.6** (0.1) | **73.5** (0.1) |
| NB-ASR | DAGNN (Thost & Chen, 2021) | 29.5 (3.9) | 40.9 (2.4) | 45.2 (1.3) | 44.0 (0.4) |
| | TA-GATES (Ning et al., 2022) | 34.0 (2.3) | 41.4 (2.0) | 44.9 (2.2) | 50.9 (0.8) |
| | FlowerFormer (Hwang et al., 2024) | 31.1 (8.0) | 44.0 (0.9) | 47.3 (1.3) | 52.2 (1.4) |
| | TEFormer (ours) | **34.8** (1.2) | **47.5** (0.5) | **49.4** (0.8) | **53.9** (0.6) |

Table 4: Comparison with state-of-the-art NAS methods on ImageNet. Search cost is measured by GPU Days (G·D). **Bold** indicates the best.

Table 5: Comparison with state-of-the-art NAS methods on CIFAR-10. Search cost is measured by GPU Days (G·D).

| Methods | Top-1/5 (%) | # P (M) | Cost (G·D) |
|---|---|---|---|
| DARTS (Liu et al., 2019) | 73.3/91.3 | **4.7** | 4 |
| DARTS+PT (Wang et al., 2021a) | 74.5/92.0 | 4.6 | 0.8 |
| Shapley-NAS (Xiao et al., 2022) | 75.7/92.7 | 5.1 | 0.3 |
| SWD-NAS (Xue et al., 2024) | 75.5/92.4 | 6.3 | 0.13 |
| TE-NAS (Chen et al., 2021) | 75.5/92.5 | 5.4 | 0.17 |
| $\xi$-GSNR (Sun et al., 2023) | 75.5/92.5 | 5.5 | 0.017 |
| SWAP-NAS (Peng et al., 2024) | 76.0/92.4 | 5.8 | **0.006** |
| AngleLoss (Yang et al., 2023) | 75.9/92.9 | 5.9 | 0.11 |
| NAO (Luo et al., 2018) | 74.3/91.8 | 11.35 | 200 |
| BONAS-D (Shi et al., 2020) | 74.6/92.0 | 4.8 | 10.0 |
| PRE-NAS (Peng et al., 2022) | 76.0/92.6 | 6.2 | 0.6 |
| TNASP-B (Lu et al., 2021) | 75.5/92.5 | 5.1 | 0.3 |
| PINAT-T (Lu et al., 2023) | 75.1/92.5 | 5.2 | 0.3 |
| CARL (Ji et al., 2025) | 76.1/92.8 | 5.3 | 0.3 |
| HyperNAS (Lv et al., 2025) | 75.4/92.5 | 6.6 | - |
| TEFormer (ours) | **76.4/93.0** | 5.6 | 0.3 |

| Methods | Acc. (%) | # P (M) | Cost (G·D) |
|---|---|---|---|
| DARTS (Liu et al., 2019) | 97.24 | 3.3 | 4 |
| DARTS+PT (Wang et al., 2021a) | 97.52 | 3.3 | 0.8 |
| Shapley-NAS (Xiao et al., 2022) | 97.57 | 3.6 | 0.3 |
| SWD-NAS (Xue et al., 2024) | 97.49 | 3.2 | 0.13 |
| $\xi$-GSNR (Sun et al., 2023) | 97.53 | 3.7 | 0.01 |
| AngleLoss (Yang et al., 2023) | 97.44 | 3.2 | 0.09 |
| NAS-BOWL (Ru et al., 2021) | 97.50 | 3.7 | 3 |
| BANANAS (White et al., 2021) | 97.43 | 3.6 | 11.8 |
| TNASP (Lu et al., 2021) | 97.48 | 3.6 | 0.3 |
| NPENAS-NP (Wei et al., 2022) | 97.56 | 3.5 | 1.8 |
| NAR-Former (Yi et al., 2023a) | 97.52 | 3.8 | 0.24 |
| PINAT (Lu et al., 2023) | 97.58 | 3.6 | 0.3 |
| CARL (Ji et al., 2025) | 97.67 | 3.7 | 0.25 |
| HyperNAS (Lv et al., 2025) | 97.61 | 3.8 | 0.1 |
| TEFormer (ours) | 97.59 | 4.0 | 0.3 |

101, 1.1 on NAS-Bench-201, and 2.7 on NAS-Bench-301 when using 1% training data. Besides, TEFormer achieves a higher Kendall's Tau than several baseline methods with significantly fewer training samples. For example, TEFormer trained with 1% data outperforms NAR-Former (Yi et al., 2023a) with 10% training data by 3.4 on NAS-Bench-101. The comparison in terms of Precision@K is reported in Table 2, showing that TEFormer ranks first on most data splits. These results clearly demonstrate the superior ranking capability of TEFormer, particularly in data-scarce scenarios.

**Results on Other Tasks:** To validate the scalability of TEFormer beyond computer vision tasks, we conduct experiments on two search spaces that focus on graph learning and automatic speech recognition tasks. As shown in Table 3, TEFormer still outperforms competitive baselines on both search spaces. While TA-GATES (Ning et al., 2022) ranks second on NAS-Bench-ASR with 1% training data, it performs poorly on NAS-Bench-Graph due to its limited ability to model the complex topology of GNNs. Instead, TEFormer leverages both local and global topological information of architectures, showing strong generalization to challenging tasks across different domains.

## 5.2 SEARCH EXPERIMENTS

Following the convention (Wen et al., 2020; Liu et al., 2022), we randomly sample a large number of architectures in DARTS (Liu et al., 2019) search space and evaluate them with the predictor. The predicted top-1 architecture is retrained to get the final test accuracy. To reduce the search cost, we train TEFormer using 100 random architectures in DARTS and their validation accuracy at 100 training epoch. The final architecture is searched on CIFAR-10 and the transferred to ImageNet. Detailed information about the searched results (mean value, standard deviation, and visualization) can be in Appendix E.5.

**Results on ImageNet:** We compare the experimental results of TEFormer on ImageNet with one-shot, zero-shot, and predictor-based NAS methods (top to bottom) in Table 4. It can be observed that TEFormer achieves a 76.4% Top-1 accuracy and a 93.0% Top-5 accuracy, outperforming other strong competitors. Although zero-shot methods like SWAP-NAS (Peng et al., 2024) take fewer search costs, TEFormer still has an advantage in classification accuracy.

**Results on CIFAR-10:** The results on CIFAR-10 are included in Table 5. Compared with other state-of-the-art methods, the proposed TEFormer still achieves competitive performance. For instance, the test accuracy of TEFormer is 0.07% higher than NAR-Former (Yi et al., 2023a) and 0.11% higher than TNASP (Lu et al., 2021), which also use Transformer as the backbone.

Table 6: Ablation studies on components of TEFormer on NAS-Bench-101. The results are scaled up by a factor of 100. **Bold** indicates the best. 'PE' is short for position encoding.

| Row | Setup | Kendall's Tau |
|---|---|---|
| 1 | TEFormer (ours) | **78.9** (3.8) |
| 2 | Remove our structural encoding | 75.3 (3.4) |
| 3 | Our structural encoding → Laplacian matrix (Lu et al., 2021) | 76.7 (3.5) |
| 4 | Our structural encoding → NAR-Former PE (Yi et al., 2023a) | 74.6 (2.6) |
| 5 | Our attention → Transformer attention (Vaswani et al., 2017) | 75.9 (2.9) |
| 6 | Our attention → FlowerFormer attention (Hwang et al., 2024) | 76.2 (2.4) |
| 7 | Remove our augmentation | 75.7 (2.8) |
| 8 | Dual similarities → Representation similarity | 77.6 (3.1) |
| 9 | Dual similarities → Label similarity | 78.3 (3.5) |
| 10 | Dual similarities → Random interpolation | 75.8 (3.4) |
| 11 | Our augmentation → NAR-Former augmentation (Yi et al., 2023a) | 75.9 (3.1) |

Table 7: Sensitivity of TEFormer to step $k$. The results are scaled up by a factor of 100.

| Step $k$ | 1 | 2 | 3 | 4 | 5 | 6 | 7 |
|---|---|---|---|---|---|---|---|
| Mean | 76.3 | 77.9 | 77.9 | 77.2 | 78.2 | **78.9** | 78.5 |
| Std | 4.7 | 4.2 | 4.4 | 4.1 | 4.2 | **3.8** | 4.0 |

Table 8: Sensitivity of TEFormer to $\alpha$. The results are scaled up by a factor of 100.

| $\alpha$ | 0.1 | 0.2 | 0.5 | 1 | 1.5 | 2 | 5 |
|---|---|---|---|---|---|---|---|
| Mean | 78.3 | 78.1 | 78.1 | **78.9** | 78.2 | 77.8 | 77.6 |
| Std | 4.2 | 4.2 | 4.1 | **3.8** | 4.1 | 4.4 | 4.7 |

## 5.3 ABLATION STUDY AND SENSITIVITY ANALYSES

To investigate the impact of each component of TEFormer, we perform ablation studies and sensitivity analyses on NAS-Bench-101 with 1% training portion. Results are averaged over 9 runs.

**Structural Encoding.** Row (2) in Table 6 demonstrates that removing structural encoding from TEFormer leads to a significant performance drop. Besides, Rows (3) and (4) reveal that the structural encoding used in previous predictors performs poorly. The superb performance in Row (1) confirms the importance of incorporating our learnable structural encoding based on random walks.

**Global Attention.** Rows (5) and (6) in Table 6 show that replacing hierarchical attention with alternative mechanisms deteriorates the performance of TEFormer, indicating the significance of modeling both intra-flow and inter-flow interactions within the architecture.

**Augmentation Strategy.** Rows (7) to (11) in Table 6 demonstrate that removing or altering the augmentation strategy causes a notable performance decline. The failure of permutation-based strategies is due to the fixed execution order in our topology-aware flow module.

**Number of Step $k$.** As shown in Table 7, the performance of TEFormer peaks at $k = 6$ and other values of $k$ also perform well. $k = 6$ yields the best results because it is equal to the depth of the architecture, which can fully express high-order topological characteristics.

**Hyperparameter $\alpha$ in Equation (10).** Table 8 shows that TEFormer is insensitive to the choice of $\alpha$. This is because interpolated samples are generated from highly similar architectures. Hence, the augmentation remains effective without much tuning of $\alpha$.

## 6 CONCLUSION

In this paper, we propose TEFormer, a novel topology-enhanced Transformer tailored for architecture performance prediction. TEFormer effectively captures local and global topological information inherent in architectures through a topology-aware flow encoding module and a hierarchical attention mechanism. Additionally, we introduce an architecture augmentation strategy that yields high-quality augmented samples to further enhance the generalization capability of TEFormer. Rigorous experiments demonstrate that TEFormer achieves state-of-the-art performance across a variety of NAS search spaces.

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

APPENDIX

In the appendix, we first provide the basic information of NAS search spaces used in our experiments in Section A. We then provide further explanations of the proposed hierarchical attention and augmentation strategy in Section B and Section C. In Section D, we provide the hyperparameter settings of TEFormer. In Section E, we present more experimental results of TEFormer.

## A    SEARCH SPACES

In this section, we will present basic information about the six search spaces.

**NAS-Bench-101** (Ying et al., 2019) contains over 423K architectures with at most seven nodes and nine edges in each architecture cell. Following the settings in FlowerFormer (Hwang et al., 2024), we use a subset of 14,580 architectures with their accuracy on the CIFAR-10 (Krizhevsky et al., 2009) dataset in our experiments.

**NAS-Bench-201** (Dong & Yang, 2020) includes 15,625 architectures with four nodes and six edges in each architecture cell. We use all the architectures with their accuracy on the CIFAR-10 dataset in our experiments.

**NAS-Bench-301** (Zela et al., 2022) is a surrogate search space with $10^{21}$ architectures. Each architecture is composed of a normal cell and a reduction cell, which has seven nodes and eight edges in each cell. Following the settings in FlowerFormer (Hwang et al., 2024), we only use a subset of 56,968 anchor architectures that are fully trained and their accuracy on the CIFAR-10 dataset in our experiments.

**NAS-Bench-Graph** (Qin et al., 2022) contains 26,206 architectures designed for graph learning tasks. The maximum number of nodes and edges in each cell is set to six and eight, respectively. We use all the architectures with their accuracy on the Cora (Sen et al., 2008) dataset in our experiments.

**NAS-Bench-ASR** (Mehrotra et al., 2021) consists of 8,242 architectures that focus on automatic speech recognition tasks. The maximum number of nodes and edges in each cell is set to four and six, respectively. We use all the architectures with their accuracy on the TIMIT (Garofolo et al., 1993) dataset in our experiments.

**DARTS** (Liu et al., 2019) is a open-domain search space that contains around $10^{18}$ architectures without available performance. Similar to NAS-Bench-301, each architecture consists of a normal cell and a reduction cell. There exist seven nodes and eight edges in each cell.

NAS-Bench-101 and NAS-Bench-Graph belong to operation-on-node (OON) search spaces, while others are operation-on-edge (OOE) search spaces. For the uniform encoding on all search spaces, we transform OOE search spaces into the OON format.

We follow the data splits in FlowerFormer (Hwang et al., 2024) in ranking experiments. For the first five search spaces, except for NAS-Bench-301, half of the architecture-performance pairs are employed as the training set and the other half are utilized as the test set. For NAS-Bench-301, the training size is 5,896 and the test size is 51,072.

## B    FURTHER EXPLANATIONS OF OUR HIERARCHICAL ATTENTION.

The hierarchical attention in TEFormer captures global topological information by combining two types of attention with a customized mask matrix: intra-flow and inter-flow attention.

(1) Intra-flow attention. This type of attention expands the receptive field of each node from its immediate neighbors to all reachable nodes along the directed computation flow. For example, if there exists a directed flow such as $v_1 \rightarrow v_2 \rightarrow v_3 \rightarrow v_4$, each node attends to all other nodes within the same flow when computing attention scores. In this way, TEFormer captures long-range global topological features along individual computation flows.

(2) Inter-flow attention. This type of attention enables information exchange across parallel computation flows within the architecture. For example, in an architecture composed of two computation flows: $input \rightarrow v_1 \rightarrow v_2 \rightarrow output$, $input \rightarrow v_3 \rightarrow v_4 \rightarrow output$, node $v_1$ and $v_3$, though

in different flows, attend to each other during attention computation because they belong to the same topological group. Hence, TEFormer captures informative global topological features across multiple computation flows.

Together, these two attention mechanisms enable TEFormer to comprehensively model both intra-flow and inter-flow dependencies of neural architectures, ensuring its capability to fully capture global topological features.

## C  FURTHER EXPLANATIONS OF OUR ARCHITECTURE AUGMENTATION.

Since previous works (Yao et al., 2022; Zha et al., 2023) have shown that random interpolation can lead to incorrect labels in non-classification tasks, we generate augmented samples by interpolating architectures that are similar at both the representation and label levels. We provide further explanations for the necessity of these similarity constraints through several examples from NAS-Bench-201.

First, representation-level similarity alone is insufficient because the representation space does not always preserve label smoothness. Specifically, architectures may be embedded close to each other in a learned representation space, yet they may yield significantly different performance. As shown in Figure 2, on NAS-Bench-201, two architectures that differ by only a single operation are highly similar at the representation level. However, their test accuracy on CIFAR-10 differs by 1.65%, which is a considerable gap for such a simple classification task. Consequently, interpolating between such representation-close but label-divergent architectures can result in augmented samples with unreliable labels, ultimately impairing the training of performance predictors.

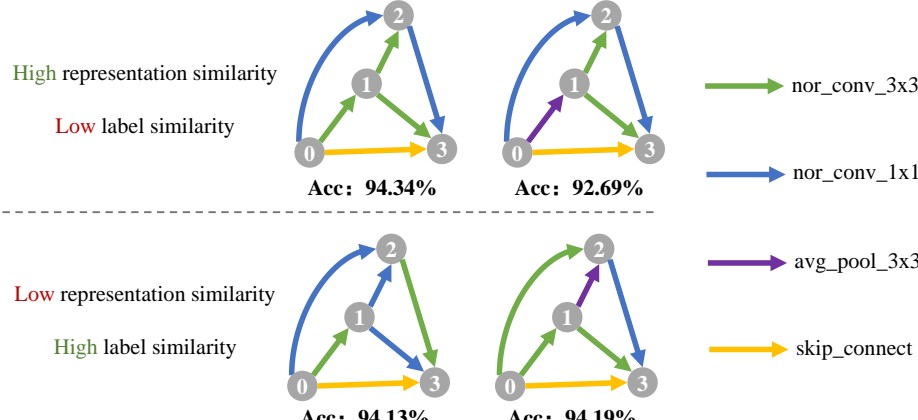

Figure 2: Motivating examples of the representation-level and label-level similarity in our interpolation-based architecture augmentation. On NAS-Bench-201, the top two architectures differ by a single operation and are highly similar in representation, yet their test accuracy differs significantly. Conversely, the bottom two architectures achieve nearly identical test accuracy but exhibit substantial structural and representational differences. The test accuracy is evaluated on CIFAR-10.

On the other hand, label-level similarity alone is also inadequate. Architectures with nearly identical performance may differ substantially in structure, and thus in representation. Figure 2 illustrates that, on NAS-Bench-201, two architectures with only a 0.06% difference in test accuracy on CIFAR-10 have very different structures, leading to low representation similarity. Interpolating between such architectures can yield invalid or out-of-distribution architectures, with unreliable interpolated labels. This may introduce noise during the training of the performance predictor and degrade its generalizability.

To ensure the quality of the augmented samples, we perform interpolation on architecture pairs that are similar at both the representation and label levels.

As for the validity of interpolated samples, we ensure this by performing interpolation in the latent space and imposing strict similarity constraints on source architectures. First, interpolation is per-

formed at the feature level in the latent space, rather than directly modifying the architecture structures. This avoids the risk of generating multiple structurally invalid neural architectures. Second, we constrain the selection to architecture pairs that are similar in both representations and performance. This ensures that interpolation occurs within a smooth and semantically meaningful region of the latent space, corresponding to valid architectures. Together, these two factors contribute to the semantic validity of the interpolated samples.

## D EXPERIMENTAL DETAILS

All the experiments are conducted on a single NVIDIA 3090 GPU. We run the ranking experiments, ablation studies, and sensitivity analyses for nine runs. As for search experiments, we run five runs on CIFAR-10 and three runs on ImageNet. The AdamW (Loshchilov & Hutter, 2019) optimizer is used to train the predictor. Following the convention (Lu et al., 2021; 2023; Yi et al., 2023a), we only report the best accuracy of the searched architecture in Tables 4 and 5 in the main text. We provide the key hyperparameters of TEFormer on six search spaces in Table 9.

Table 9: Hyperparameters of TEFormer on NAS-Bench-101, NAS-Bench-201, NAS-Bench-301, NAS-Bench-Graph, NAS-Bench-ASR, and DARTS.

| Hyperparameters | NB-101 | NB-201 | NB-301 | NB-G | NB-ASR | DARTS |
|---|---|---|---|---|---|---|
| Encoder Layers | 10 | 16 | 8 | 4 | 6 | 8 |
| Hidden Dim | 64 | 64 | 256 | 256 | 128 | 256 |
| Attention Heads | 16 | 8 | 8 | 8 | 8 | 8 |
| Attention Dropout | 0.5 | 0.5 | 0.3 | 0.4 | 0.1 | 0.3 |
| Random Walk Step $k$ | 7 | 4 | 6 | 3 | 4 | 6 |
| Ranking Loss Margin $b$ | 0.5 | 0.5 | 0.1 | 0.005 | 0.1 | 0.1 |
| Augmentation Coefficient $\lambda_1$ | 0.2 | 0.2 | 0.3 | 0.6 | 0.6 | 0.3 |
| Interpolation Intensity $\alpha$ | 1 | 1 | 1 | 1 | 1 | 1 |
| Batch Size | 32 | 64 | 64 | 16 | 64 | 64 |
| Learning Rate | 8.0e-4 | 5.0e-4 | 3.0e-3 | 7.0e-3 | 9.0e-4 | 3.0e-3 |
| Weight Decay | 4.0e-5 | 6.0e-5 | 2.0e-5 | 4.5e-6 | 1.8e-3 | 2.0e-5 |
| Epochs | 200 | 400 | 200 | 100 | 100 | 200 |

## E ADDITIONAL EXPERIMENTAL RESULTS

### E.1 COMPUTATIONAL EFFICIENCY AND ANALYSIS

We conduct a study to compare the computational efficiency of TEFormer with three state-of-the-art predictors: NAR-Former (Yi et al., 2023a), TA-GATES (Ning et al., 2022), and Flower-Former (Hwang et al., 2024). For a fair comparison, we train all the predictors on 1% training data of NAS-Bench-101 using a single NVIDIA 3090 GPU. The batch size is set to 128, and the number of epoch is set to 200.

**Efficiency results**. As shown in Table 10, TEFormer takes around 3× less training time than NAR-Former, which also uses a Transformer backbone. Besides, TEFormer requires comparable training and inference time to TA-GATES and FlowerFormer but achieves consistent performance improvements over them. Since the main computational cost of training the predictor lies in the acquisition of labeled training samples, the minor difference between TEFormer and FlowerFormer in training time can be ignored.

**Analysis**. The computational cost of TEFormer increases modestly with architecture size and number of operations. The cost primarily arises from two components: topology-aware flow encoding whose cost scales linearly, and hierarchical attention whose cost scales quadratically with architecture size. Denote the number of operations and edges as $N$ and $E$. Topology-aware flow encoding traverses all nodes bidirectionally along directed edges. The forward and backward passes each

Table 10: Training/inference time, and model parameter on NAS-Bench-101 with 1% training data.

| Predictors | Training time (secs) | Inference time (secs) | # Params |
|---|---|---|---|
| NARFormer (Yi et al., 2023a) | 301.54 (15.88) | 5.87 (0.19) | 4,882,081 |
| TA-GATES (Ning et al., 2022) | 81.98 (2.76) | 6.19 (0.33) | 348,065 |
| FlowerFormer (Hwang et al., 2024) | 77.38 (3.08) | 6.05 (0.21) | 901,459 |
| TEFormer (ours) | 85.00 (3.17) | 6.29 (0.16) | 1,003,101 |

incur a cost of $O(E)$, which scales approximately linearly with architecture size. Hierarchical attention incurs a worst-case cost of $O(N^2)$ when the attention mask is fully connected. However, in practice, the cost is lower due to the sparsity introduced by our hierarchical attention. This component scales approximately quadratically. As most search spaces use small-sized cells (fewer than 10 operations and 15 edges), TEFormer remains computationally efficient.

The attention mask matrix computation and storage introduce no significant computational or memory overhead. For computation, the mask matrix can be pre-computed before training, avoiding repeated computation during runtime. For storage, the mask matrix has the same size as the adjacency matrix of the architecture and directly replaces it in our implementation, so it does not incur any extra memory cost.

Table 11: Comparison with state-of-the-art NAS methods on ImageNet. **Bold** indicates the best.

| Predictors | Top-1 Acc. (%) | MACs |
|---|---|---|
| OFA (Cai et al., 2020) | 76.0 | 230M |
| NARFormer (Yi et al., 2023a) | 76.9 | 571M |
| PINAT (Lu et al., 2023) | 77.8 | 452M |
| TEFormer (ours) | **78.6** | 491M |

### E.2 SEARCHING ON MOBILENET SPACE

We conduct experiments to search for architectures on a larger MobileNet space ($> 10^{19}$ architectures) in the real-world mobile setting (MACs<600M). Specifically, we strictly follow OFA's (Cai et al., 2020) settings except for replacing the original predictor with TEFormer. Table 11 shows that TEFormer can discover the architecture with higher accuracy on ImageNet compared with strong NAS baselines, demonstrating its practicability on larger search spaces and real-world tasks.

### E.3 CROSS-SPACE PREDICTIONS

To validate the effectiveness of TEFormer in cross-space scenarios, we strictly follow the CDP's (Liu et al., 2022) evaluation procedure and only replace the original GCN backbone with TEFormer. TEFormer is trained on NAS-Bench-101 and NAS-Bench-201, and estimated on the shallow DARTS space. As shown in Table 12, TEFormer achieves superior performance on the unseen search space without any retraining, highlighting its strong generalization capability in cross-space settings.

Table 12: Kendall's Tau on Shallow DARTS space. The results are scaled up by a factor of 100. **Bold** indicates the best.

| Predictors | Kendall's Tau |
|---|---|
| CDP (Liu et al., 2022) | 53.06 |
| TEFormer (ours) | **55.67** |

### E.4 ABLATION ON THE AUGMENTATION LOSS

We investigate the contribution of the proposed augmentation loss, Equation(12), to TEFormer on five search spaces with 1% training portion. As shown in Table 13, removing the augmentation loss brings consistent performance drops, demonstrating that the proposed augmentation loss generalizes well across multiple tasks.

Table 13: Ablation studies of our augmentation loss on NAS-Bench-101, NAS-Bench-201, NAS-Bench-301, NAS-Bench-Graph, and NAS-Bench-ASR. Results are scaled up by a factor of 100. **Bold** indicates the best.

| Kendall's Tau | NB-101 | NB-201 | NB-301 | NB-Graph | NB-ASR |
|---|---|---|---|---|---|
| with loss in Equation(12) | **78.9** | **81.1** | **66.9** | **52.0** | **34.8** |
| without loss in Equation(12) | 75.7 | 80.2 | 65.5 | 50.9 | 34.3 |

Table 14: Ablation for our augmentation strategy on ImageNet using MobileNet space. **Bold** indicates the best.

| Predictors | Top-1 Acc. (%) | MACs |
|---|---|---|
| TEFormer (wo/ our augmentation) | 76.4 | 378M |
| TEFormer (w/ our augmentation) | **78.6** | 491M |

We also conduct a comprehensive evaluation on ablation results in a chain-based space (MobileNet space) and the number of augmentation samples. For the experiments on the chain-based MobileNet space (Cai et al., 2020). we replace the original predictor in OFA with TEFormer and the ablation variant (without augmentation). The results in Table 14 show that our augmentation strategy helps find architectures with higher top-1 accuracy on ImageNet, demonstrating the effectiveness of our augmentation in chain-based search spaces.

Table 15: Effects of number of augmented samples. The results are scaled up by a factor of 100.

| Augmentation number | 1 | 4 | 8 | 16 | 32 |
|---|---|---|---|---|---|
| Mean | 78.9 | **79.5** | 79.1 | 76.7 | 75.8 |
| Std | **3.8** | 4.6 | 5.3 | 6.3 | 7.9 |

For the experiments on the number of augmentation samples, we increase the number of augmentation samples up to the batch size and select samples according to their similarity scores in descending order. Kendall's Tau on NAS-Bench-101 with 1% training data is shown in Table 15. We observe that adding a small number of high-similarity augmentation samples slightly improves Kendall's Tau (78.9 → 79.5 for 1 → 4 samples). However, adding more augmentations beyond this point leads to a decrease in mean performance (79.1 → 75.8 for 8 → 32 samples) and an increase in standard deviation (5.3 → 7.9 for 8 → 32 samples), as lower-similarity samples introduce noise and reduce TEFormer's stability. This shows that using a single augmentation sample already achieves satisfactory performance.

### E.5 DETAILS OF THE SEARCHED ARCHITECTURE ON DARTS

In parallel to prior works (Liu et al., 2019; Peng et al., 2022; Lu et al., 2023), we search architectures using DARTS search space on CIFAR-10 and directly transfer the searched architecture to ImageNet (Deng et al., 2009). The average accuracy of the searched architecture on CIFAR-10 is 97.52±0.05% (97.50%, 97.45%, 97.53%, 97.53%, 97.59%). The average Top-1 and Top-5 accuracy on ImageNet is 76.3±0.13% (76.1%,76.4%,76.3%) and 93.1±0.05% (93.0%,93.1%,93.1%), respectively. The searched architecture is shown in Figure 3.

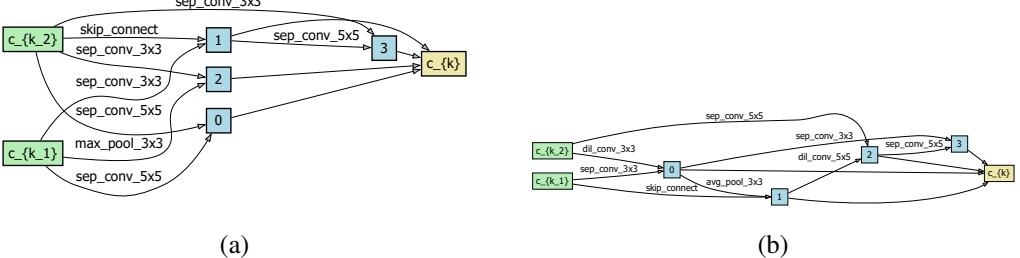

(a) (b)

Figure 3: Normal cell (*Left*) and reduction cell (*Right*) searched by TEFormer.

