# OpenReview forum: "TEFormer: A Topology-Enhanced Transformer for Architecture Performance Prediction"
_ICLR.cc/2026/Conference — Submitted to ICLR 2026_

### Official Review · Reviewer_FtPd · 2025-10-23

**Soundness:** 3
**Presentation:** 2
**Contribution:** 1
**Rating:** 4
**Confidence:** 5

**Summary:**

This paper proposes TEFormer, or Topology-Enhanced Transformer for Neural Architecture Search (NAS) performance prediction. TEFormer continues a line of work on flow-based predictors [1-3] for NAS. TEFormer augments these approaches by encoding the local and global features through a specialized attention mechanism and positional encoding. TEFormer is evaluated on several NAS-Benchmarks for computer vision and graph prediction.

**Strengths:**

- The paper makes advances in NAS performance prediction.
- The experimental setup and execution is solid.
- The method is clear and easy to understand.
- Ablation studies are provided.
- The evaluation is not simply limited to NAS-Bench-{101, 201, 301}, but other benchmarks.

**Weaknesses:**

- The method is incredibly incremental. Table 6 proves this with how little, arguably not statistically significant change takes place during the ablation study.
- TEFormer primarily stems from an existing, but very narrow line of work on Flow-based predictors [1-3] and the contribution is just slight increases in Kendall's Tau on DARTS which is not noteworthy when we've already been able to leap frog over the best DARTS architectures on ImageNet for several years [4].
- The statement in lines 224-225 "Considering that both forward and backward passes are essential for accurately modeling neural architectures (...), we encode the bidirectional computational flow", should be removed or heavily revised, as it is essentially ignoring other advances in performance prediction [5, 6, 7] that are tangential to [1-3]; it essentially reads as if the method of [1-3] is the only correct way to achieve performance prediction, which is not true.

**Questions:**

Can the local and global level features be used to extract information about the structure of good/bad architectures as [9, 10] do?

References:

[1] https://arxiv.org/abs/2004.01899

[2] https://proceedings.neurips.cc/paper_files/paper/2022/file/d0ac28b79816b51124fcc804b2496a36-Paper-Conference.pdf

[3] https://arxiv.org/abs/2403.12821

[4] https://arxiv.org/abs/1812.00332

[5] https://arxiv.org/abs/2506.04001

[6] https://arxiv.org/abs/2210.03230

[7] https://proceedings.neurips.cc/paper_files/paper/2022/hash/572aaddf9ff774f7c1cf3d0c81c7185b-Abstract-Conference.html

---

> ### Author Response · Authors · 2025-11-21
> **Thank you for your valuable comments.**
>
> We thank you for recognizing our clear methodology and comprehensive experiments. We are also grateful for your valuable feedback.
>
> >  **W1: Limited empirical improvement**.
>
> Thank you for raising this concern. We would like to clarify that our improvements are non-trivial, and the ablation results are statistically significant. As shown in Table 6, removing each key component leads to a clear drop in ranking performance on NAS-Bench-101: removing the learnable structural encoding causes a 3.6% drop, replacing our hierarchical attention with Transformer-style attention leads to a 3.0% drop, and removing the augmentation results in a 3.2% decrease. These results indicate a substantial contribution of each component. Note that the performance gap between the previous state-of-the-art FlowerFormer[1] (75.0%) and the previous runner-up DAGNN[2] (72.4%) is only 2.6% (See Table 1).
>
> Furthermore, we conducted paired t-tests between our full model and each ablated variant (removing structural encoding, switching to Transformer-style attention, and removing augmentation). The resulting p-values are 0.007, 0.017, and 0.009, respectively. All are below 0.05, confirming that the observed improvements are statistically significant and consistent.
>
> These results collectively demonstrate the substantial contribution of the proposed modules rather than incremental changes.
>
> > **W2: Limited contribution based on flow-based methods.**
>
> Thank you for your comments.
>
> For **flow-based methods**, this line of research is valuable and worth further development because it closely mimics the actual computation flow in neural networks and has achieved strong success[1-4] in recent years. However, existing works still exhibit two key limitations. First, they struggle to distinguish operations that share the same type but play different topological roles. Second, they overlook the dependencies across parallel computation flows. Our method addresses both issues through learnable structural encoding and a hierarchical attention mechanism, thereby substantially extending the capability of the flow-based paradigm.
>
> For **contributions of our method**, we would like to clarify that we did not conduct ranking experiments (measured by Kendall’s Tau) on the DARTS search space. Instead, we used DARTS solely for search experiments, where our searched architecture achieves competitive ImageNet performance of 76.4% top-1 accuracy, which is 1.3% higher than Proxyless (75.1%) that you mentioned. Moreover, as shown in Table 11 of Appendix E.2, we searched architectures on ImageNet using the MobileNet search space and obtained an even higher 78.6% top-1 accuracy. These results demonstrate that our method is capable of discovering high-performing architectures for image classification tasks across different search spaces.
>
> > **W3: Presentation about modeling bidirectional flows.**
>
> Thank you for pointing this out. We would like to clarify that the original statement was intended specifically in the context of flow-based methods. It was not meant to imply that flow-based approaches are the only correct way to perform architecture performance prediction, and we fully recognize the success of non-flow-based methods. To avoid any potential misunderstanding, we have revised the statement in the updated manuscript as follows:
> '' Considering that both forward and backward passes are included when modeling neural architectures in flow-based methods (Ning et al., 2022; Hwang et al., 2024), we follow this convention and encode the bidirectional computation flow.''
>
>
> > **Q1: Extracting the structure of good or bad architectures.**
>
> Thank you for the insightful question. Our method is designed as a general performance predictor backbone, and in principle it can be combined with additional structure-extraction techniques to analyze properties of good and bad architectures. However, the specific identities of references [9, 10] were not provided in the review, so a direct comparison or integration with their approaches is not feasible.
>
> In addition, we would like to clarify that the local and global features learned by TEFormer cannot be directly used to extract high-level structural patterns of good/bad architectures. These features are operation-level representations, capturing local and global dependencies between operations, rather than architecture-level features that quantify how much each operation contributes to the overall architecture performance.
>
> **Reference**
>
> [1] FlowerFormer: Empowering Neural Architecture Encoding using a Flow-aware Graph Transformer. In CVPR, 2024.
>
> [2] Directed Acyclic Graph Neural Networks. In ICLR, 2021.
>
> [3] A Generic Graph-based Neural Architecture Encoding Scheme for Predictor-based NAS. In ECCV, 2020.
>
> [4] TA-GATES: An Encoding Scheme for Neural Network Architectures. In NeurIPS, 2022.
>
> [5] ProxylessNAS: Direct Neural Architecture Search on Target Task and Hardware. In ICLR, 2019.

---

> > ### Comment · Reviewer_FtPd · 2025-11-26
> >
> > I thank the authors for their rebuttal. After careful consideration, I concur with reviewer mjum that the paper's weaknesses outweigh the strengths and thus elect to maintain my score.

---

### Official Review · Reviewer_mjum · 2025-10-29

**Soundness:** 2
**Presentation:** 3
**Contribution:** 2
**Rating:** 4
**Confidence:** 5

**Summary:**

This paper proposes TEFormer, a novel Transformer-based predictor for neural architecture performance estimation. The core innovations include a topology-aware flow encoding module that incorporates bidirectional computation flows with learnable structural encodings based on random walks, a hierarchical attention mechanism to model both intra-flow and inter-flow dependencies, and an interpolation-based architecture augmentation strategy to combat data scarcity. The method is evaluated on multiple NAS benchmarks across computer vision, graph learning, and speech tasks, demonstrating highly competitive ranking performance and search results.

**Strengths:**

1. The idea of explicitly modeling bidirectional computation flows (forward and backward) is well-motivated and differentiates the work from many existing predictors that rely on static graph encodings. The integration of learnable structural encodings from random walks is a principled approach to capture rich topological information.
2. The authors evaluate on many standard NAS benchmarks and provide ablation studies and sensitivity analyses, showing consistent gains and some robustness to hyperparameters.

**Weaknesses:**

1. Some design choices are not sufficiently explained, especially from Eq. (2) to Eq. (4). The rationale behind these formulations is not intuitive.
2. In Section 4.2, the underlying motivation for computing attention only between nodes connected by a directed path or within the same topological group is not well-justified. The authors' explanations read more like descriptions of the rules' effects rather than a justification of the underlying design principles.
3. The interpolation-based augmentation strategy, presented as a major contribution, appears potentially risky. A more comprehensive evaluation is required to substantiate its value, such as conducting ablations in both chain-based and cell-based search spaces and studying the impact of the number of augmented samples.
4. The experimental comparisons lack several important baselines, such as [1], [2], [3], and [4]. Specifically, a direct comparison with the transformer-based predictor [1] is missing on CIFAR-10, and on ImageNet, only the cell-based results of [1] are compared, omitting its chain-based results. Furthermore, the results of [2], [3], and [4] appear to surpass those reported in this work.
5. For the results on CIFAR-10, it is necessary to report both the mean and standard deviation.

[1] PINAT: A Permutation INvariance Augmented Transformer for NAS Predictor
[2] CARL: Causality-guided Architecture Representation Learning for an Interpretable Performance Predictor
[3] HyperNAS: Enhancing Architecture Representation for NAS Predictor via Hypernetwork
[4] Computation-friendly Graph Neural Network Design by Accumulating Knowledge on Large Language Models

**Questions:**

see Weaknesses.

---

> ### Author Response · Authors · 2025-11-21
> **[1/2] Thank you for your comments.**
>
> We sincerely thank you for the recognition of our novelty, comprehensive experiments and promising performance. We are also grateful for your constructive feedback.
>
> > **W1: Further explanations for design choices in Eq.(2) to Eq.(4).**
>
> Thank you for the valuable suggestion. Eq. (2)–(4) together describe the update rule of the forward flow encoder, where each node integrates (i) its own previous hidden state and (ii) the information propagated from all its predecessors. These equations follow a standard message-passing paradigm, but are adapted specifically for modeling computation flows in neural architectures.
>
> For Eq.(2):
> \begin{equation}
>     h_{v,u}^{l} = \mathrm{Softmax}({w_1^l}^\top{h_v^l} + {w_2^l}^\top{h_u^{l-1}}+ {w_3^l}^\top{e_{v,u}^l})\ h_v^{l},\quad v \in P(u)
> \end{equation}
>
> This equation computes the message $h^l_{v,u}$ from a predecessor node $v$ by assigning an attention weight based on the predecessor’s feature $h^l_v$​, the target node’s previous feature $h_u^{l-1}$, and the corresponding edge feature $e^l_{v,u}$, which captures operation-level relationship. (We also note that Eq. (2) contained a typo in the initial submission, which has now been corrected in the revised version.)
>
> For Eq.(3):
> \begin{equation}
>     h_{P(u)}^{l} = \mathrm{Agg}(\{h_{v,u}^{l}| v \in P(u)\})
> \end{equation}
> Eq. (3) then aggregates all predecessor messages $h^{l}_{P(u)}$ to obtain the overall incoming information for node $u$. We use simple addition as the aggregation function since it preserves the contribution of each predecessor.
>
> For Eq.(4):
> \begin{equation}
>     h_u^{l} = \mathrm{GRU}(h_u^{l-1},h_{P(u)}^{l})
> \end{equation}
> Eq. (4) performs the state update of node $u$ using a GRU. The rationale is that GRU is effective for modeling sequential dependencies and information accumulation across steps. In the context of modeling computation flows, nodes are naturally processed in a topological sequence, where each update depends on previously computed states. This results in a more stable and expressive propagation of flow information across layers.
>
> The formulas and corresponding descriptions have been revised in the updated version to improve clarity.
>
> > **W2: The underlying motivation for computing attention only between nodes connected by a directed path or within the same topological group.**
>
> Thank your for the valuable comment. The overall motivation for our hierarchical attention
> is to model the flow-based dependencies among operations. It can be divided into two types of customized attention.
>
> - **Intra-flow attention** (nodes connected by a directed path). Since the partial order in DAG-like neural architectures induces specific topological relations between connected operations that reflect the computation flow, a given operation’s predecessors and successors are naturally more important than other nodes. Therefore, we restrict attention to nodes along a directed path to model the intra-flow dependencies instead of full attention.
>
> - **Inter-flow attention** (nodes within the same topological group) Operations that fall into the same topological group are executed in parallel and process information coming from the same preceding depth. Such parallel operations often provide complementary information (e.g., multi-scale features in Inception-style architectures), and their joint behavior defines the semantic state of that topological group. Therefore, we allow attention among nodes within the same topological group .
>
> The above presentation explains the motivation behind our hierarchical attention design, highlighting why intra-flow and inter-flow dependencies are modeled.
>
> >**W3: A more comprehensive evaluation for the interpolation-based augmentation strategy.**
>
> Thanks for your valuable suggestions. We conduct a more comprehensive evaluation on
> ablation results in a chain-based space (MobileNet space) and the number of augmentation samples.
>
> - **Experiments on the chain-based MobileNet space.** We strictly follow OFA’s [1] search and evaluation settings except for replacing the original predictor with TEFormer and the ablation variant (without augmentation). The results in the table below show that our augmentation strategy helps find architectures with higher top-1 accuracy on ImageNet, demonstrating the effectiveness of our augmentation in chain-based search spaces.
>
> | Predictors                      | Top-1 Acc. (%) on ImageNet | MACs |
> | ------------------------------- | -------------------------- | ---- |
> | TEFormer (wo/ our augmentation) | 76.4                       | 378M |
> | TEFormer (w/ our augmentation)  | **78.6**                   | 491M |

---

> > ### Comment · Reviewer_mjum · 2025-11-26
> >
> > Thank you for your efforts, the detailed explanations, and the additional experiments. After carefully reviewing the responses, I still believe that many aspects of the paper’s design lack principled justification (e.g., W2 and W3), which makes the contribution appear meaningful primarily from a practical point of view. After considering the authors’ replies and the comments from the other reviewers, I have decided to maintain my score.

---

> ### Author Response · Authors · 2025-11-21
> **[2/2] Thank you for your comments.**
>
> **Experiments on the number of augmentation samples.** We increase the number of augmentation samples up to the batch size and select samples according to their similarity scores in descending order. Kendall's Tau on NAS-Bench-101 with 1% training data is shown below. We observe that adding a small number of high-similarity augmentation samples slightly improves Kendall’s Tau (78.9 → 79.5 for 1 → 4 samples). However, adding more augmentations beyond this point leads to a decrease in mean performance (79.1 → 75.8 for 8 → 32 samples) and an increase in standard deviation (5.3 → 7.9 for 8 → 32 samples), as lower-similarity samples introduce noise and reduce TEFormer’s stability. This shows that using a single augmentation sample already achieves satisfactory performance.
>
> | Number of Augmentation Sample |  1   |  4   |  8   |  16  |  32  |
> | :---------------------------: | :--: | :--: | :--: | :--: | :--: |
> |           **Mean**            | 78.9 | 79.5 | 79.1 | 76.7 | 75.8 |
> |            **Std**            | 3.8  | 4.6  | 5.3  | 6.3  | 7.9  |
>
> We have added these experiments in Appendix E.4 of the revised paper.
>
> > **W4: Experimental comparisons lack several important baselines.**
>
> Thank you for your valuable comments. We have added the results of PINAT [2], CARL [3], and HyperNAS [4] on CIFAR-10 and ImageNet in Table 4 and 5. Since DesiGNN [5] is specifically designed for GNNs, it is not applicable to CNN search, and thus we cannot provide its results for CNNs.
>
> On CIFAR-10, TEFormer (97.59%) remains competitive and only falls short of CARL (97.67%) and HyperNAS (97.61%) by 0.08% and 0.02%, respectively. While on ImageNet, TEFormer (76.4%) achieves a higher top-1 accuracy of 0.3% and 1.0% compared to CARL (76.1%) and HyperNAS (75.4%).
>
> Regarding PINAT’s results in the chain-based search space, a direct comparison with the methods based on cell-based DARTS space in Table 4 would be unfair. To ensure fairness, we compared TEFormer in a chain-based space (MobileNet space) with PINAT and added the results to Table 11 in Appendix. We observe that TEFormer (78.6%) still outperforms PINAT (77.8%) in this setting.
>
> We have added these results in the revised paper.
>
> > **W5: The mean and standard deviation of results on CIFAR-10.**
>
> Thank you for your suggestion. The mean and standard deviation of the results on CIFAR-10 are 97.52% ± 0.05%. The accuracies over five independent runs are 97.50%, 97.45%, 97.53%, 97.53%, and 97.59%, respectively. These results were already included in Appendix E.5 of the original paper, and we now highlight them in Section 5.2 of the revised paper.
>
>
> **References**
>
> [1] Once-For-All: Train One Network And Specialize it For Efficient Deployment. In ICLR, 2020.
>
> [2] PINAT: A Permutation INvariance Augmented Transformer for NAS Predictor. In AAAI, 2023.
>
> [3] CARL: Causality-guided Architecture Representation Learning for an Interpretable Performance Predictor. In ICCV, 2025.
>
> [4] HyperNAS: Enhancing Architecture Representation for NAS Predictor via Hypernetwork. In Arxiv, 2025.
>
> [5] Computation-friendly Graph Neural Network Design by Accumulating Knowledge on Large Language. In Arxiv, 2024.

---

### Official Review · Reviewer_eGKW · 2025-10-29

**Soundness:** 3
**Presentation:** 3
**Contribution:** 2
**Rating:** 4
**Confidence:** 3

**Summary:**

TEFormer is a novel Transformer model designed for NAS performance prediction. It precisely captures the complex topological information of neural architectures by combining Topology-aware Flow Encoding and a Hierarchical Attention Mechanism. Additionally, the model employs an interpolation-based augmentation strategy in the latent space to enhance generalization in few-shot scenarios. TEFormer achieves SOTA performance across multiple NAS benchmarks and includes detailed ablation studies confirming the effectiveness of its components.

**Strengths:**

1.  The paper provides extensive experimental validation.
2.  The paper includes detailed sensitivity analysis and ablation studies.

**Weaknesses:**

1.Overall, the core of this paper's contribution is essentially the introduction of a new loss function, which is derived by combining several typical neural network modules, thus lacking novelty.
2. The paper lacks corresponding theoretical proof. I believe providing a relevant theoretical analysis for the proposed loss function would have been a significant addition to this paper.

**Questions:**

Can you try to theoretically analyze why this network architecture was chosen?

---

> ### Author Response · Authors · 2025-11-21
> **Part-1. Thanks for your comments.**
>
> Thank you for recognizing our detailed sensitivity analysis, ablation studies, and comprehensive experiments. We are also grateful for your valuable feedback.
>
> >  **W1: The core contribution is a new loss function and the novelty of this paper.**
>
> Thank you for raising this concern. We would like to clarify that the core contribution of our paper is not the loss function. The loss only corresponds to the interpolation-based augmentation component, which serves to improve generalization during few-shot training.
>
> The main contributions of our method lie in three novel and complementary designs:
>
> (1) **A learnable structural encoding** based on random walks, capturing long-range local topology more effectively than previous designs relying on adjacency matrices, node depths, or computation flows.
>
> (2) **A hierarchical attention mechanism** that selectively models intra-flow and inter-flow dependencies within architectures to incorporate global topological features, offering advantages over fully connected or only intra-flow attention schemes used in prior work.
>
> (3) **An interpolation-based architecture augmentation strategy** that greatly enriches training data beyond the simple operation permutation widely used in previous methods.
>
> These innovations are tightly coupled to mutually enhance TEFormer's ability to capture topological characteristics of neural architectures. This is not a simple stacking of standard modules. Instead, their combination is carefully designed to address limitations of prior flow-based predictors. Specifically, the learnable structural encoding captures **local dependencies** between operations, the hierarchical attention mechanism models **global dependencies** across computation flows, and the interpolation-based architecture augmentation strategy mitigates potential **overfitting** of the employed graph Transformer backbone in few-shot scenarios like architecture performance prediction.
>
> Therefore, the novelty of our work lies in these customized and complementary designs, rather than in a single loss function.

---

> ### Author Response · Authors · 2025-11-21
> **Part-2. Thanks for your comments.**
>
> > **W2: Theoretical analysis for the loss function.**
>
> We thank the reviewer for this insightful comment. Below we show that **incorporating augmented pairs that satisfy both feature similarity and label similarity constraints yields a smaller generalization error bound than training without augmentation** under the assumption of using a two layer neural architecture with ReLU function.
>
> **Preliminary**. Given an input pair $(x,y),(x',y') \in (\mathcal{X} × \mathcal{Y})^2$, where $\mathcal{X}\in \mathbb{R}^d$ is the input space and $\mathcal{Y}\in \mathbb{R}$ is the label space, let $f:\mathcal{X} \rightarrow \mathbb{R}$ be the ranking function on $\mathcal{X}$ and $l:\mathbb{R}\times (\mathcal{X}\times\mathcal{Y})^{2}\rightarrow \\{0,\mathbb{R}_{+}\\}$ be the ranking loss function. According to prior works[1][2], we define the expected error of $f$ as:
>
> $$
>           \quad\quad     R_{l}(f) = \mathbb{E}_{(X,Y),(X',Y') \sim (\mathcal{X}\times\mathcal{Y})^{2}}\left[l(f,(X,Y),(X',Y')) \right.]  \tag{1}
> $$
>
> Given a training dataset $\mathcal{D}=\{(x_i,y_i)\}_{i=1}^{N} \subseteq (\mathcal{X},\mathcal{Y})^N$, the empirical ranking risk can be formulated as:
>
> $$
>  \quad\quad  \widehat{R}_{l}(f) = \frac{1}{N(N-1)}
> \sum _ {i=1} ^ {N-1} \sum _ {j=i+1} ^ {N}
> l\big(f,(x_i,y_i),(x_j,y_j)\big) \tag{2}
> $$
>
> The regularized empirical error is:
>
> $$
> \quad\quad \widehat{R} _ {l}^{\lambda}(f)
> = \widehat{R} _ {l}(f) + \lambda C(f). \tag{3}
> $$
> where the last term is a regularization term and $\lambda$ > 0.
> Our Hinge ranking loss $l_{rank}$ (the first part of our overall loss) can be reformulated as:
>
> $$
> \quad\quad l _ {rank} =
> \left[ b - \big( f(x) - f(x') \big)\cdot \operatorname{sign}(y - y')
> \right] _ + \tag{3}
> $$
>
> where $[·] _ + = max(0, ·)$ and $b$ is a margin parameter.
>
> Next, we provide the generalization error bound of hinge ranking loss $l_{rank}$.
>
> **Theorem 1**. Let $\mathcal{A}$ be a symmetric ranking algorithm whose output on the dataset
> $\mathcal{D}\in(\mathcal{X}\times\mathcal{Y})^N$ is $f_{\mathcal{D}}=\arg\min _ {f\in\mathcal{F}} \widehat{R}^{\lambda} _ {l}(f)$,  where $N$ is the size of training set.  Assume the inputs and hypothesis class satisfy  $\|x\|\le c_x$ for all $x\in\mathcal{X}$ and $\|f\| _ 2 \le c_f$ for all $f\in\mathcal{F}$.  Assume the hinge ranking loss satisfies $0\le l_{rank}(f,(x,y),(x',y'))\le L$ . Then for any $0 < \delta < 1$, with probability at least $1-\delta$, we have:
>
> $$
>  \quad\quad R_{l_{rank}}(f_{\mathcal{D}})< \widehat{R} _ {l_{rank}}(f_{\mathcal{D}}) + \frac{8c_x^{2} c_f^{2}}{\lambda N} + \left(\frac{4c_x^{2} c_f^{2}}{\lambda} + L\right) \sqrt{\frac{2\ln(1/\delta)}{N}}. \tag{4}
> $$
>
> Detailed Proof for Theorem 1 can be found in Eq(1) to Eq(9) in Appendix of ReNAS[2].  Note that Eq.(4) can already represent the generalization error bound of **our loss without augmentation**.
>
> Next, we give the generalization error bound of **our overall loss** $l_o = l_{rank} + \lambda_1 l_{aug}$, where $\lambda_1>0$, $l_{aug}$ denotes the augmentation loss which is same as $l_{rank}$ except that it computes each augmented sample and its corresponding source samples.
>
> We first compute pairwise difference bound for augmented samples. Each augmented sample $(\tilde x,\tilde y)$ is generated by interpolation $\tilde x=\alpha x_i+(1-\alpha)x_j,\ \tilde y=\alpha y_i+(1-\alpha)y_j$ for some $α∈[0,1]$, where the source pair $(x_i,y_i),(x_j,y_j)$ satisfies $x_i^\mathrm{T} x_j\geq \tau_x,|y_i-y_j|\le \tau_y.$ In practice, the interpolation is performed at the representation level rather than directly on architectures; this formulation is for theoretical analysis. Under the similarity constraints above, for each augmented sample:
>
> $$\quad\quad |x_i - x_j| \le \sqrt{2(1-\tau_x)}, \quad |y_i - y_j| \le \tau_y \tag{5}$$
>
> Assuming the ranking function $f$  is $L_f$​-Lipschitz in the input space, we use the 1-Lipschitz property of hinge function $[\cdot] _ +$​, for any augmented pair $(\tilde x, \tilde y)$ and source pair $(x_j, y_j)$, $(x_i,y_i)$, we have:
>
> $$\quad\quad \big| l_{\text{rank}}(f, (x_i, y_i), (\tilde x, \tilde y)) - l_{\text{rank}}(f, (x_j, y_j), (\tilde x, \tilde y)) \big| \le \big| (f(\tilde x) - f(x_i)) \cdot \operatorname{sign}(\tilde y - y_i) - (f(\tilde x) - f(x_j)) \cdot \operatorname{sign}(\tilde y - y_j) \big|$$
> $$
> \quad\quad \le | f(x_i) - f(x_j) | + | y_i - y_j | \\ \le L_f \| x_i - x_j \| + |y_i - y_j| \le L_f \sqrt{2(1-\tau_x)} + \tau_y \tag{6}
> $$
>
> The number of original samples is $N$, giving $N(N−1)/2$ original pairs. Let the number of augmented samples be $S$, providing 2$S$ pairs (each augmented sample forms two pairs with source samples). The number of overall data pairs is $N(N−1)/2+2S$. So the effective sample number can be roughly calculated as $N'$, which is larger than $N$.

---

> ### Author Response · Authors · 2025-11-21
> **Part-3. Thanks for your comments.**
>
> Hence, the generalization error bound of our overall loss can be obtained in a similar way to Eq.(4).  Assume the hinge ranking loss satisfies $0\le l_{rank}(f,(x,y),(x',y'))\le L$ . Then for any $0 < \delta < 1$, with probability at least $1-\delta$, we have:
>
> $$
> \quad\quad R _ {l_o}(f_{\mathcal{D}}) < \widehat{R} _ {l_o}(f_{\mathcal{D}}) + \lambda_1 \epsilon_{\text{sim}} + \frac{8 c_x^2 c_f^2}{\lambda N'} + \left( \frac{4 c_x^2 c_f^2}{\lambda} + L \right) \sqrt{\frac{2 \ln(1/\delta)}{N'}}. \tag{7}
> $$
>
> where $\epsilon_{\text{sim}} = L_f \sqrt{2(1-\tau_x)} + {\tau_y}$. Since the source pairs of augmented samples are highly similar, we have $\tau_x \approx 1$ and $\tau_y \approx 0$. As a result, the second term on the right-hand side is negligible. Moreover, because $1/N' < 1/N$, **incorporating augmented pairs that satisfy both feature similarity and label similarity constraints (Eq.(7)) leads to a tighter generalization error bound compared to training without augmentation (Eq.(4)).**
>
>
> > **Q1: Theoretical analysis for the architecture design.**
>
> Thank you for your valuable suggestion. Since we provide a detailed theoretical analysis of the loss functions in our response to W2, here we focus on the other two core components of TEFormer, both of which are designed to address the challenges of modeling topology in flow-based performance predictors.
>
> - **Learnable structural encoding**.  We use a learnable random walk-based encoding to capture local relationships among operations. Unlike adjacency matrices, node depths, or fixed computation flows, this encoding can model **long-range local topology** precisely. Each node’s representation incorporates multi-hop structural context, providing richer and more flexible local features than prior methods.
>  - **Hierarchical attention mechanism.** We employ a hierarchical attention mechanism to model both intra-flow and inter-flow dependencies through a tailored attention mask. Intra-flow attention captures interactions along the individual flow, while inter-flow attention aggregates global relationships across different computation flows. This **global modeling** avoids the noise of fully connected attention and the narrow focus of intra-flow-only schemes.
>
> The learnable structural encoding captures expressive **local** topological features, while the hierarchical attention incorporates rich yet non-redundant **global** topological information. Together, they enable effective learning of architectural representation from both local and global perspectives, greatly enhancing the model’s stability and generalization.
>
> **References**
>
> [1] Generalization Bounds for Ranking Algorithms via Algorithmic Stability. In JMLR, 2009.
>
> [2] ReNAS: Relativistic Evaluation of Neural Architecture Search. In CVPR, 2021.

---

### Meta-Review · Area_Chair_TMWB · 2025-12-23

**Summary:**

The paper proposes TEFormer, a Transformer-based performance predictor for Neural Architecture Search (NAS). The method introduces three main components: (1) a topology-aware flow encoding module that uses learnable structural encodings based on random walks to capture local topology; (2) a hierarchical attention mechanism that models intra-flow and inter-flow dependencies ; and (3) an interpolation-based architecture augmentation strategy to improve generalization in data-scarce scenarios. The authors demonstrate that TEFormer achieves state-of-the-art performance on benchmarks like NAS-Bench-101/201/301 and ImageNet, outperforming recent baselines.

Despite the strong empirical results and the authors' effort to address specific questions during the rebuttal, the AC’s recommendation is Reject. The consensus among reviewers is that the contribution is incrementally novel, largely representing an engineering improvement over prior flow-based predictors (specifically FlowerFormer) rather than a fundamental conceptual advance. While the authors successfully provided requested theoretical analysis and additional baselines, the reviewers remained unconvinced by the principled justification for the specific architectural design choices.

**Reviewer Concerns:**

**Addressed Concerns:**
- Missing Baselines: Reviewer mjum requested comparisons against PINAT, CARL, and HyperNAS. The authors successfully added these comparisons, showing TEFormer outperforms PINAT and is competitive with CARL and HyperNAS on CIFAR-10 and ImageNet.
- Theoretical Analysis: Reviewer eGKW criticized the lack of theoretical proof for the loss function. The authors provided a detailed generalization error bound analysis based on algorithmic stability theory in their rebuttal.
- Augmentation Validity: Reviewer mjum questioned the risk and generalizability of the interpolation strategy. The authors addressed this by adding ablation studies on a chain-based search space (MobileNet), demonstrating the strategy works beyond cell-based spaces.

**Outstanding Concerns:**
- Incremental Novelty: Reviewer FtPd and Reviewer eGKW maintained that the work is an incremental combination of existing modules (standard flow-based methods + Transformer attention) rather than a significant innovation. Reviewer FtPd specifically noted that the gains over FlowerFormer, while statistically significant, are small and the conceptual delta is narrow.
- Principled Design Justification: Reviewer mjum remained unconvinced by the justification for the specific design choices (e.g., Eq. 2-4 and the specific attention masks). Even after the rebuttal explanation, the reviewer felt the design lacked "principled justification" and appeared to be meaningful primarily from a practical, heuristic perspective rather than a theoretical one.

**Reviewer Scores:**

- Reviewer eGKW: (4 to 5). While the reviewer asked for theoretical analysis which was provided 17171717, their fundamental concern about the "core of this paper... lacking novelty" likely prevents a strong positive shift.
- Reviewer mjum: (4 to 4). The reviewer explicitly stated post-rebuttal that they decided to "maintain [their] score" because the design choices still lacked principled justification.
- Reviewer FtPd: (4 to 4). The reviewer explicitly stated post-rebuttal that the "weaknesses outweigh the strengths" regarding the incremental nature of the work and elected to maintain the score.

---

### Decision · Program_Chairs · 2026-01-26

Reject